# Does Robotic Assisted Technique Improve Patient Utility in Total Knee Arthroplasty? A Comparative Retrospective Cohort Study

**DOI:** 10.3390/healthcare12161650

**Published:** 2024-08-19

**Authors:** Matteo Ratti, Daniele Ceriotti, Riccardo Rescinito, Rabia Bibi, Massimiliano Panella

**Affiliations:** 1Department of Translational Medicine (DiMeT), Università del Piemonte Orientale, 28100 Novara, Italy; 10036607@studenti.uniupo.it (D.C.); 10033325@studenti.uniupo.it (R.R.); 20036124@studenti.uniupo.it (R.B.); massimiliano.panella@uniupo.it (M.P.); 2Habilita S.p.A., Casa di Cura Villa Igea, Str. Moirano, 2, 15011 Acqui Terme, Italy

**Keywords:** utility, robotic surgery, knee surgery, RATKA

## Abstract

Background: Even if robotic assisted total knee arthroplasty (RATKA) is a widely used technique, there is still a gap of knowledge about whether this technology is effective in improving the patient utility. This measure is of paramount importance for conducting cost effectiveness analysis. The aim of this study was to compare the utility measure derived from self-reported outcomes questionnaires in patients who underwent RATKA compared to patients who underwent the manual surgery. Methods: We compared 72 patients operated with a robotic technique with 70 operated with traditional technique. The utility data were collected with the WOMAC (Western Ontario and McMaster University Osteoarthritis index) self-administrated questionnaire that investigates pain, stiffness and functionality of the patients, an then mapped to a utility value through a validated transforming function. We performed three surveys: the first one before the intervention (t0), the second one 1 year after the surgery (t1) and the third one at the 2 year follow up (t2). Results: we observed higher utility values in both groups. In detail, the mean utility score in the RATKA group increased from 0.37 to 0.71 (t1) and 0.78 (t2), while in the conventional group it increased from 0.41 to 0.78 (t1) and 0.78 (t2). The fixed effect coefficients of t1 and t2 were found to be 0.37 and 0.363 (*p* < 0.001 for both). The coefficient of the robotic technique, along with its interaction with the t1 and t2 time effect was non-significant. Conclusions: Even if at t1 the utility of patient who underwent RATKA were lower, at longer follow up (t2) we found no significant difference compared to traditional technique, leaving the superiority of robotic assisted technique yet to be proved. Our results may be useful for calculating the gained or lost Quality Adjusted Life Years (QALYs), so that the health care system (or an insurance company) could make an appropriate decision whether to fund the robotic approach or not, after a careful assessment of the incremental costs incurred.

## 1. Introduction

Total knee arthroplasty (TKA) is an elective surgical procedure for end-stage knee osteoarthritis. Each year, over 700,000 TKAs are performed in the US and more than 100,000 in the UK [1,2]. In Italy, the national arthroplasty registry reports 82,828 total knee replacements performed in 2019, with growing trends since the early 2000s [1,3,4]. In this context the health outcomes of the TKA remain suboptimal: in fact, approximately 20% of the operated patients currently report dissatisfaction because of residual limited function and persistent pain even years after the intervention [5,6]. Since these symptoms are often reputed to be associated with a soft tissue imbalance or a misalignment of the prosthetic implant components, the robotic-assisted total knee arthroplasty (RATKA) has recently gained attention as a possible approach to achieve better clinical outcomes [7,8]. In fact, some recent studies proved that RATKA is related to increased accuracy regardless of the surgeon’s experience and demonstrated excellent early post operative health outcomes. [9,10] A recent systematic review analyzing 2113 knees in 13 studies reported that no difference in functional outcomes was detected and recommended more robust evidence for widespread adoption. [11] The early surgery robots used automatic technologies that provided limited possibility to adapt the intervention to patient’s unexpected intraoperative needs. Therefore, a new generation of semi automatic devices was designed with the goal to make the procedure more customizable while reaching higher precision both in component alignment and in soft-tissue balance [12,13]. The literature about the topic is highly fragmented; moreover, the majority of the studies concern the automatic models of robot, which are now discontinued [14,15]. The retrieved studies adopt different methodologies and heterogeneous outcome measures, predominantly radiological rather than clinical, with the patient and public health perspectives often left overlooked [16]. While the former have the advantages of a hard endpoint for a scientific study, the ultimate choice to fund the RATKA technology relies on the utility values of the patients combined with the incremental costs that the national health care system (or a private insurance company) may want to pay. Therefore, a deep understanding of the utility values, representing the preference of the patient, is necessary and yet poorly described in literature. Moreover, in Italy many regional local health authorities are providing an extra reimbursment for robotic techniques for different interventions, which might not yet be supported by evidence. For this reason we investigated the utility values of the patients who underwent both manual TKA and RATKA interventions through an observational cohort study.

## 2. Materials and Methods

### 2.1. Study Design and Setting

We performed a retrospective cohort study comparing robotic surgery with traditional surgery. Patients were recruited at “Habilita Casa di Cura—Villa Igea” center (Acqui Terme, Alessandria, Italy) from September 2020 to August 2021. We studied two groups of patients: one group underwent RATKA and the other group performed conventional TKA using manual instruments. The records of eligible patients operated with TKA were collected, anonymized and subsequently analyzed with their consent. Patients completed the WOMAC (Western Ontario and McMaster University Osteoarthritis index) questionnaire before the surgery and both at the 1 year and 2 year follow up, and then the utility values were derived through a mapping funcion. The ethical committee of Alessandria approved the study with protocol number 2268 on 27 January 2022.

### 2.2. Population

Inclusion criteria consisted of patients with an age between 65 and 80 years who were suffering from severe osteoarthritis (with radiographic evidence of structural damage). Moreover, they were required not to be responsive to conservative therapy for at least 3–6 months, with a significant impact on quality of life (uncontrolled, moderate, or severe stiffness, pain and limited functionality measured with a PROM), and with a preoperative WOMAC of at least 20. Instead, we excluded patients with cognitive impairment or for whom the PROM measurement methods failed to detect outcomes, patients with an ASA (American Society of Anesthesiology) score more than 3, and patients who had surgery on the controlateral knee in the year before the observation period of the study. We extracted the following characteristics: age, gender, BMI (Body Mass Index) and ASA score.

### 2.3. Conventional Surgery Operative Details

The conventional surgery employed a medial parapatellar approach. After a vertical anterior incision, the patella is everted and dislocated laterally. The knee is then flexed to 90° to obtain the exposure of the entire joint allowing the surgeon to manually conduct both the proximal (femoral) and the distal (tibial) resections. Finally, he cementates the implant components and assesses their alignment with a standard radiography along with the manual assessment of the soft tissue balance.

### 2.4. RATKA Technique Operative Details

The ROSA^®^ Knee System (Zimmer Biomet, Warsaw, IN, USA) is a robotic platform whose objective is to assist orthopedic surgeons with the arthroplasty interventions. It also helps assessing the condition of the soft tissues to facilitate implant position. Moreover, the surgeon can use a planning software preoperatively in order to design implant positioning and sizing with high accuracy [17]. The device is composed of two units positioned at the two sides of the operating table. The first one is a unit that consists in two parts: a compact robotic arm and a touchscreen; the second one is an optical unit and a further touchscreen. The robotic arm is equipped with a force sensor that allows the surgeon to move the robotic unit manually to the desired position by measurement of forces exerted at the end of the arm and a compensation principle, regardless of the implant used [18,19].

### 2.5. Utility and Outcomes Measures

The patient completed the WOMAC questionnaire, a validated and commonly used PROM (Patients Reported Outcome Measure) at three times: preoperatively (t0), at the 1 year follow up (t1), and finally at the 2 year follow up (t2). It is a self-administered questionnaire that consists of 24 questions grouped in three domains: pain (5 questions), stiffness (2 questions), and physical function (17 questions). Each question can be answered on a five-point Likert scale (none, mild, moderate, severe, and extreme) that was scored from 0 to 4. The final score for WOMAC was calculated by adding the scores of the three categories and normalizing the total value to 100. The final score was therefore ranging from 0 (healthy subject) to 100 (severe impairment of pain/stiffness and physical function). In our study, we labeled the categories into pain (WOMAC P), stiffness (WOMAC S) and functionality (WOMAC F) and we adopted the Italian version of the questionnaire. The WOMAC score of each of the three domains was then mapped to the utility value through a validated beta regression function outlined from the study of Bilbao and coll. [20].

### 2.6. Sample Size

Being the utility a derived endpoint, the study was designed to test the difference in the postoperative WOMAC score between the two groups. To detect a difference of 10 points (value of the minimal clinical importance difference [21]) relative to the mean preoperative value a minimum sample of 62 patients per group was calculated.In fact, a recent systematic review confirmed that improvement of the WOMAC less than 10 points are not linked to an higher satisfaction of the patients, even if there are inconsistences among the values retrieved [22]. The sample size assessment set a power of study of 80% and an alpha level of 0.05. We also considered an attrition rate of 10% and performed a two tail *t*-test. The calculation was made with the pwr package of R.

### 2.7. Statistical Analysis

The quantitative variables were described with mean and standard deviation, while the qualitative one were reported with percentages and proportions. Considering the normal distribution of the WOMAC scores and the utility derived value, a two-tails *t*-test was used for the continuous measures, whereas Chi-square and Fisher’s exact test were applied to the proportional measures. We also tested a generalized linear mixed model (GLMM), with the patient factor modeled as a random effect. The data analysis was carried out with R 4.4.0 with R Studio 2023.09.1 Build 494 (packages: lme4, tidyverse, tableone) and statistical significance was considered at *p* values of less than 0.05.

## 3. Results

An overall of 142 TKA consecutive clinical records of the considered period (September 2020-August 2021) were analaysed: 72 RATKA patients and 70 operated with manual technique. The two groups were similar for age, gender, BMI and ASA score (Table 1). The mean age was 68.7 years for the robot group and 68.8 years for the conventional surgery group. The proportion of female individuals was 60% and the mean BMI was roughly 28.5 for both groups. All the preoperative WOMAC scores of the two groups were almost identical. In particular, the total score in the RATKA group was 65.12 (SD: 17.21), while it was 61.77 (SD: 18.96) in the other group. We observed a dropout rate of 0% at t1 20% at t2.

At the 1 year follow up, we observed higher utility scores in both groups. In the RATKA group, the total mean score rose from 0.37 to 0.71, while in the other group we observed an increase from 0.41 to 0.78. The utility value difference between groups at this timepoint was found to be statistically significant, in favor of an higher value for the conventional group. At the 2 year follow-up the utility value was found to be 0.78, regardless of the technique employed. Figure 1 shows the boxplots of the improvements in utility values after the intervention, for both of the surgical techniques.

Table 2 reports the GLMM coefficients of the fixed effects. The model reported statistically significant values for the utility values at t1 (+0.370) and t2 (+0.363) when compared to t0, while detecting no interaction between them and the surgical technique.

## 4. Discussion

In our study, both the RATKA and conventional TKA groups showed a significant improvement in their utility scores. Therefore, both techniques seem to be effective in reaching satisfactory results for the majority of patients, despite the reported 20% dissatisfaction rate. However, the difference between the utility values at t1 favours the traditional technique, suggesting that the improvement in RATKA may be a little slower. Notably, even if the mean values were comparable, the boxplot revealed that the median utility value of the RATKA group was lower than the traditional group. Our opinion is therefore cautious about this finding, which may depend on the distribution of the variable. Interestingly, two patients (one per group) reported a negative pre-operatory utility, thereby valuing this state as worse than dead; their utility however improved to 0.42 and 0.90 at the 1 year follow up (t1).

Some studies showed better clinical outcome scores when applying RATKA technology, implying an higher utility value with respect to the traditional one. For instance, Marchand et al. [23] compared 53 consecutive robotic-arm-assisted (RAA) patients to 53 consecutive manual TKAs with a one-year follow-up period and observed a better total WOMAC score with robotic cohort (6 points vs. 9; *p* < 0.05). Another study by Yang et al. [24] observed a decrease in postoperative WOMAC score after a long term (10 years) follow-up of 71 individuals in a robotic cohort in comparison to 42 conventional TKA (7.6 vs. 11.5; *p* = 0.12). However, it is worth to note that these studies employed a robot of an older generation, so we reasonably think that these results may not be valid anymore nor fully comparable with ours. Our study showed different results, leaving the superiority of RATKA yet to be demonstrated, at least concerning the patient utility. Moreover, the relative bigger improvement of the utility at t1 of the patients who underwent traditional technique suggests more resarch to be focused on early health outcomes.

We also noticed that the RATKA literature is quite heterogeneous and we were unable to retrieve any study concerning the utility values. In fact, a recent systematic review by Agarwal et al. [15] analysed four different outcomes: the Hospital Special Surgery score (HSS), the Knee Society Score (KSS), the Range Of Motion (ROM) and the WOMAC score, but with no attempt to derive an utility measure. The authors concluded that RATKA is not convincingly superior to conventional surgery since two of these measures (KSS and ROM) yielded no statistically significant improvements. Another review of outcomes by Shatrov et al. [25] also reported a small improvement of PROMS but no strong conclusion about RATKA. A recent 52 patient study found no improvement of functional outcomes, but no increase in blood loss or complications, therefore supporting only the safety of the robotic assistance [26].

The current national health care system reimbursement tariff in Italy for this kind of intervention (Principal Diagnosis ICD-9-CM: 71516, Principal OR intervention: 8154, resulting in DRG number 544) is 8.837 euros. Even if the Regional authorities have the opportunity to increase the amount in case of robotic technology employment, such as the majority of them did for prostatectomy, to our knowledge no additional revenue has been approved yet. Our results suggest that a specific health technology assessment (HTA) study is mandatory before reaching such a decision, because the only ministerial document is dated 2017 and predominantly included gynecology and urology disciplines, because of the high cost of those peculiar robots (DaVinci) [27]. However, a recent study claimed that the cost savings derived from the RATKA are those concerning the utilization of home care services [28], which are difficult to quantify in a health care system like the Italian one, which keeps the care of chronic conditions distinct from the acute ones (the hospitals and the home care services belong to different public companies). For these reasons we think that a thorough assessment of the utility values (and the relative quality-adjusted life years) connected to both the TKA and the RATKA represent an urgent need and the first step towards evaluating economic sustainability.

Our study presents some limitations. First, being derived only from a self reported outcome measure, the results are heavily dependent on what has been reported by the patients. In fact, although of wide use, the WOMAC is known to suffer from persisting problems such as different normalizations of the scores [29]. In addition, even though we considered a meaningful number of potential confounders such as age, BMI, gender, ASA and preoperative PROM score, we cannot be sure that our results have not been conditioned by other unknown factors. Another limitation to the generalizability of our results is due to the enrollment of individual whose mean BMI is well over 25. We cannot therefore exclude that results could significantly differ in a normal 19–25 BMI population.

Despite the interventions with RATKA and standard techniques were made by the same surgical team, we don’t have enough data to set a learning curve that describe the level of training. However, the team had more than one year expertise in intervention with the ROSA system, so we reasonably think that the learning curve have already had reached the plateau at the time of the first considered interventions. In addition, we don’t have the information about the alignment of the components, so we don’t know if the desired relative position was achieved. However, we think that this fact is not important for our conclusions, because we were focused on the utility values only.

Although these limitations, our study showed that there is still a gap in knowledge of utility value, not allowing a proper evaluation of the QALY gained or lost using this technique. Since this technology is relatively new and rapidly changing, we think that urgent further high quality studies would be helpful to assess whether the robotic technology represents a true improvement of the TKA surgery.

In brief, based on our results we agree with the findings of the systematic reviews by Agarwal and Chin [15,30] that it is still not clear whether RATKA represents a clear improvement for the patient suffering from knee osteoarthritis.

## 5. Conclusions

Based on our results, we found that at early follow up conventional technique still yielded more favourable results when considering the patient utility. However, on a longer timespan, this differences disappeared, leaving the superiority of the robotic technique yet to be proved. Nevertheless, our results may be useful for calculating the gained or lost QALYs, so that the health care system (or an insurance company) could make an appropriate decision whether to fund the robotic approach or not, after a careful assessment of the incremental costs incurred.

## Figures and Tables

**Figure 1 healthcare-12-01650-f001:**
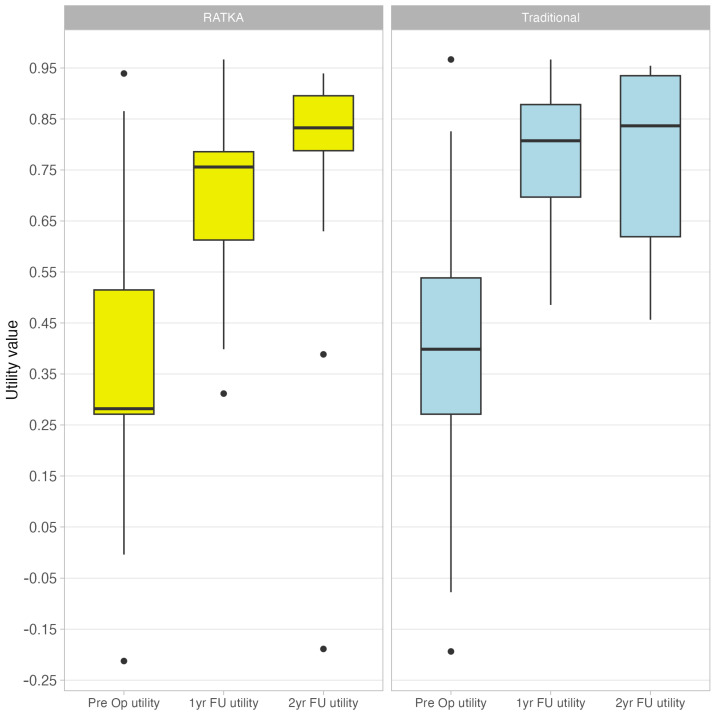
Utility values at t0, t1 and t2 for both RATKA and conventional group.

**Table 1 healthcare-12-01650-t001:** Patients characteristics. All differences were tested with two-tails independent samples *t*-test or Chi-Square test as appropriate.

	Overall	Traditional Technique	Robotic Assisted	*p*
n	142	70	72	
Gender = M (%)	56 (39.40)	28 (40.00)	28 (38.90)	1
Asa Score (%)				0.945
1	31 (22.00)	16 (23.20)	15 (20.80)	
2	81 (57.40)	39 (56.50)	42 (58.30)	
3	29 (20.60)	14 (20.30)	15 (20.80)	
Discharge type = at home(%)	62 (44.00)	34 (49.30)	28 (38.90)	0.284
Age (mean (SD))	68.72 (10.11)	68.77 (9.54)	68.67 (10.70)	0.951
BMI (mean (SD))	28.36 (4.57)	28.43 (4.56)	28.29 (4.61)	0.858
Pre-operatory Womac Pain Score (mean (SD))	62.57 (19.36)	60.86 (20.11)	64.24 (18.59)	0.3
Pre-operatory Womac Stiffness Score (mean (SD))	66.99 (20.59)	64.64 (22.62)	69.27 (18.28)	0.182
Pre-operatory Womac Functionality Score (mean (SD))	63.32 (19.25)	61.70 (19.66)	64.89 (18.85)	0.326
Pre-operatory Womac Overall Score (mean (SD))	63.47 (18.11)	61.77 (18.96)	65.12 (17.21)	0.272
Pre-operatory (t0) utility value (mean (SD))	0.39 (0.22)	0.41 (0.22)	0.37 (0.21)	0.316
1 year (t1) utility value (mean (SD))	0.74 (0.13)	0.78 (0.11)	0.71 (0.15)	0.001
2 year (t2) utility value (mean (SD))	0.78 (0.22)	0.78 (0.19)	0.78 (0.24)	0.979

**Table 2 healthcare-12-01650-t002:** GLMM model results of fixed effect coefficients.

Fixed Effect	Ref.	Estimate	Std. Err.	t	*p* Value
(Intercept)		0.418	0.068	6.188	<0.001
Robotic assisted surgery	Traditional technique	−0.036	0.031	−1.175	0.241
Male gender	Female	0.029	0.021	1.366	0.174
Bmi		−0.001	0.002	−0.293	0.77
1 yr utility value (time effect)	Pre-op utility	0.370	0.031	11.931	<0.001
2 yr utility value (time effect)	Pre-op utility	0.363	0.054	6.732	<0.001
Interaction: Robotic surgery—utility at t1 (1 yr)		−0.036	0.044	−0.820	0.413
Interaction: Robotic surgery—utility at t2 (2 yr)		0.042	0.070	0.604	0.546

## Data Availability

Anonymized dataset could be provided upon motivated request.

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
