# Peer review of "Does Robotic Assisted Technique Improve Patient Utility in Total Knee Arthroplasty? A Comparative Retrospective Cohort Study"

_healthcare, 2024, doi:10.3390/healthcare12161650_

Round 1

Reviewer 1 Report

Comments and Suggestions for Authors

The authors provided a good manuscript on "Patient Utility after Robotic Total Knee Arthroplasty: a Comparative Retrospective Cohort Study". 

There are some main points that should be improved before publication:

The title can be improved and rewritten using more details.

The conclusion of the abstract does not supported the all results.

The introduction is short. it can be improved using more recent and relevant references.

the sample size determination needs some more details.

Please uniform the digits for Tables.

The conclusion section must be improved. It does not support the results.

Author Response

Dear Reviewer, thank you for your valuable advices, please refer to the attached file for a point to point reply.

Kind Regards, 

mr

Reviewer 2 Report

Comments and Suggestions for Authors

This study provides a comprehensive analysis between robotic-assisted total knee arthroplasty (RATKA) and traditional manual surgery, focusing on patient utility values derived from the WOMAC questionnaire​​. The study emphasizes the necessity for additional rigorous research to ascertain whether RATKA offers a substantial advancement over conventional procedures, particularly in view of the swift progress of robotic technologies.

The reviewer believes that providing an overview of the components and tools used in RATKA components (such as the Knee System's robotic arm, optical unit, and preoperative planning software, or others), in the introduction will enhance the reader's understanding of the technology. 

Despite the apparent sufficiency of the sample size, it is imperative to provide a more comprehensive justification of potential dropouts and other factors that influence statistical power in order to improve the robustness of the results.

Comments on the Quality of English Language

readable

Author Response

Dear Reviewer, we thank you for your time used in evaluating our work. Please refer to the attached file for a point-by-point response.

Kind Regards, 

mr

Reviewer 3 Report

Comments and Suggestions for Authors

Dear authors,

A well written paper and conducted study.

Results section

In Figure 1, please remove the background grid lines.

Discussion section

The second paragraph in the discussion only refers to previous studies. Please update the second paragraph to include a contrast and comparison with your current study. 

Kindest regards

Author Response

Dear Reviewer,

thak you for your time passed in evaluating our work. Please refer to the attached file for a point by point response.

Kind Regards,

mr
